### Common characteristics of directional spreading-steepness joint 1 distribution in freak wave events 2 3 S. H. Liu<sup>1±</sup>, Y. Z. Li<sup>2</sup>, and X. Y. Yue<sup>1</sup> 4 <sup>1</sup>National Marine Data & Information Service, Tianjin, China 5 <sup>2</sup>Applied Ocean Physic and Engineering, Woods Hole Oceanographic Institution, Woods Hole, MA, USA 6 7 Abstract 8 Seven freak wave incidents previously documented in the real ocean in combination with model 9 hindcast simulations are used to study the variations associated with freak wave-related 10 parameters, such as wave steepness, directional spreading, and frequency bandwidth. Unlike the 11 strong correlations between the freak wave parameters and freak waves' occurrence which were 12 obtained in experimental and physical research, the correlations are not clear in the freak waves 13 occurredoccurring in the real ocean. Wave directional spreading-steepness joint distribution is 14 introduced and common visual features were found in the joint distribution when freak waves 15 occur among seven "freakish" sea states. The visual features show that freak wave incidents occur 16 when the steepness is large and directional spreading is small. Besides the large steepness is large 17 and <u>small</u> directional spreading is small, a long-duration relatively rough sea state is also 18 necessary for the freak wave generation. The joint distribution is more informative than the 19 sequential variation of any single statistical wave parameter. The continuous sea states of local 20 large steepness and small directional spreading are supposed to be "freakish" sea statesgenerate 21 freak waves, and two-dimensional distribution visualization is found to be a useful tool for freak 22 waves forecast. The common visual features of joint distributions supply an important cue for the

- 23 theoretical and experimental research.
- 24

## 25 1 Introduction

Freak wave (also known as rogue wave, extreme wave, and unexpected wave) has been a hot 27 topic during the last decades in engineering and science research. Recently, two candidate 28 mechanisms that lead to freak waves are debated. One is linear and the other is nonlinear. The 29 linear mechanism is considered as to be a result of linear focusing in fixed time and position due 30 to ocean wave's dispersion, geometrical, current and wind force (Kharif and Pelinovsky, 2003). 31 Nevertheless, freak wave is essentially a nonlinear phenomenon because of the large wave 32 steepness of freak waves. Freak waves could also be produced as a result of the instability of 33 ocean waves. Because of the abrupt and huge energy focusing characteristics of freak waves, the 34 instability is more considered to be self-instability rather than externally forced. Benjamin and 35 Feir (1967) found the instability of uniformly traveling trains of Stokes waves, the Benjamin-Feir 36 instability (B-F instability). B-F instability is considered as the most probable candidate for the 37 freak wave occurrence, which has been validated by lots of experimental and physical results. The 38 studies on freak waves' dynamics are mostly focused on the B-F instability and the extreme wave 39 events can be caused by B-F instability in different various circumstances.

From the engineering point of view, the experimental and theoretical achievements should be
validated in the ocean and be applied in practice. Its vV alidation is difficult due to the rareness of

1

freak waves and insufficient large-scale measurements. Most of the in-situ observations of freak 43 waves are time-series surface elevation measurements, which can not provide spatial and 44 directional spectrum information. There are some efforts that aim to set up a freak wave 45 early-warning system in the ocean by experimental and theoretical research (Janssen, 2003; Mori 46 and Janssen, 2006; Mori et al., 2011; Akhmediev et al., 2011a, b). Recent research found that 47 some wave parameters have high correlation with freak waves' occurrence. Under unidirectional 48 or small directional spreading (long-crested) conditions, the probability of freak waves is 49 considered to increase when wave steepness increase and spectrum narrows (Gramstad and 50 Trulsen, 2007; Waseda et al., 2009; Onorato et al., 2010). According to the results of hindcast 51 simulated "freakish" sea states, it is expected to find the conditions that trigger freak waves in the 52 ocean and check if the theoretical and experimental achievements findings are also applicable to 53 oceanic freak waves. It will This check gives useful information of certain circumstances which 54 trigger freak waves and complement existing theoretical framework of freak waves.

# 55 2 Model configurations

As a state-of-the-art third generation spectral model, WAVEWATCH III (WW3) (Tolman, 56 57 2002, 2009) offers good descriptions of statistical sea states from a kinetic approach that well 58 mimics the directional spectrum. Although the WW3 model can not give the simulation of simulate 59 freak waves, the freak-wave related parameters deduced from simulated results can be considered 60 as an approximation of corresponding parameters of statistical sea states which is pertinent to 61 freak waves. Short-lived freak waves can last only for a few wave periods (Janssen, 2003) and 62 hardly influence relatively long-time wave statistical characteristics (Toffoli and Bitner-Gregersen, 63 2011). Even in complex conditions, the evolution of the spectrum within the spectral-kinetic 64 description appears to be consistent both qualitatively and quantitatively with solutions for the 65 weakly nonlinear dynamical equations for ocean waves (Zakharov et al., 2007; Badulin et al., 2008). 66 67 Seven freak wave incidents in the ocean used in this study and the defined model grid are 68 shown in Table 1. Hindcast simulations are conducted by WW3 multi-grid technique. The simulated results are easily affected by the errors propagated from the outside boundary of model 69 70 grid, so the inner grids that cover the freak wave incidents' positions are set in the middle of outer

grids, so the finite grids that cover the freak wave medents positions are set in the initial of outer grids. The coarse resolution for outer grid is  $0.25 \times 0.25$  and the fine resolution for the inner is  $0.1 \times 0.1 \times 10^{-1}$ . The implementations of WW3 in our simulations use the default model setting as defined in Tolman (2002, 2009) with few exceptions. The wave directions are <u>set-resolved</u> to <del>36</del>-

74 <u>10° (by 10 degree36 "bins"</u>); and the number of frequencies ranges from 0.0412 to 0.4056 in 25
 <u>bandsis set to 25 levels</u>, with the increment factor of 1.1. The freak wave incidents do not occur in

the shallow water, so only three source terms are considered in the model: wind-wave interaction

term, nonlinear wave-wave interactions term and a dissipation (whitecapping) term. We force the

78 <u>wave model</u> useing the Cross-Calibrated, Multi-Platform Ocean Surface Wind Velocity (Atlas et

al., 2011) to force the wave model, which ishas 0.25 degree resolution at 6 hours interval. A

80 reanalysis ocean current from National Marine Data & Information Service (China) is also taken

into account in the model for the diagnosis of the results. The nonlinear wave-wave interaction
 term is calculated by high resolution DIA method (Tolman, 2002). For WW3, it always needs 1 to

2 days to spin-up the model in "cold start" conditions. In our simulations, we all-allow more than 3

84 days for the model to spin-up before the freak wave incident time.

Table1. Time, position information and model set up of freak wave incidents

| Case    | Time(UTC)         | Position    | Outer grid of model | Inner grid of model | Note      |
|---------|-------------------|-------------|---------------------|---------------------|-----------|
| Case1   | 30 Dec 1980 05:30 | 156°11'E,   | 115°-180°E,         | 140°-160°E,         | Northwest |
|         |                   | 31º N       | 10°-65°N            | 25°-40°N            | Pacific   |
| Case2   | 23 Jun 2008 04:00 | 144°-145°E, | 115°-180°E,         | 140°-160°E,         | Northwest |
|         |                   | 35°-36°N    | 10°-65°N            | 25°-40°N            | Pacific   |
| Case3   | 13 Dec 1978 00:00 | 44°N, 24°E  | 70°W-10°E,          | 30°W-20°W,          | Atlantic  |
|         |                   |             | 10°-75°N            | 40°-50°N            |           |
| Case4   | 1 Jan 1995 15:20  | 2°28'E,     | 70°W-10°E,          | 5°W-5°E,            | New Year  |
|         |                   | 58°11'N     | 10°-75°N            | 55°-65°N            | Wave      |
| Case5.1 | 18 Nov 1997 01:10 | 1º44'E,     | 70°W-10°E,          | 5°W-5°E,            | Alwyn oil |
| Case5.2 | 20 Nov 1997 01:51 | 60°45'N     | 10°-75°N            | 55°-65°N            | platform  |
| Case6   | 27 Jul 2002 12:00 | 22.17°E,    | 0.5°E-60°E,         | 17°E-27°E,          | FA        |
|         |                   | 37.97 °S    | 70°S-0°N            | 43°S-33°S           | platform  |

case1, case2 and case3 are for ship sinkings which are thought to be caused by freak waves.

case4, case5 and case6 are freak waves that are recorded by in-situ measurements.

#### 3 Results and discussion 88

Seven hindcast simulations are aimed to obtain the directional spectrum that covers time span 89 90 for the freak waves. Statistical wave parameters, including significant wave height (Hs), wave

steepness (  $\delta$  ), directional spreading (  $\sigma_{_{ heta}}$  ), frequency peakedness (  $Q_{_p}$  ) and BFI (the ratio

between steepness and spectral bandwidth) are derived from directional spectrum. The 92

Hs,  $\delta$  ,  $\sigma_{\theta}$  are defined following Tolman (2002).  $Q_p$ , BFI (Eqs. 1 and 2) are defined as Janssen 93

and Bidlot (2003). We seek to check the parameters that sethaving a close relationship with freak 95 wave occurrence and find physically-meaningful factors common to "freakish" sea states.

$$Q_{P} = 2m_{0}^{-2} \int_{0}^{\infty} \sigma \left[ \int_{0}^{2\pi} F(\sigma, \theta) d\theta \right]^{2} d\sigma$$
(1)

$$BFI = k_o m_o^{1/2} Q_p \sqrt{2\pi}$$
(2)

where  $\sigma$  is the relative radian frequency,  $\theta$  is the wave direction,  $k_a$  is the wave number,

F is the wave energy density spectrum,  $m_o$  is the zero order moment of F.

Hs is an important parameter that characterizes the mean sea states. It always takes a local

extreme value (case1, case3, and case6) or near-the\_extreme value when freak waves occur (Fig.

1). Many in-situ observations have demonstrated that the freak wave occurrence will increase

significantly in quite rough seas (Guedes et al., 2003; Liu et al., 2009), so the quasi local extreme 104 value feature is self-consistent to some extent. Case 5 indicates the that freak wave events may

occur when the Hs are not the highest locally in continuous time series, unlike the other s'-cases'

quasi local extreme value feature (Fig. 1, case5). This means freak waves can also take place 106

### 107 relatively far away from local extreme sea states.

Steepness, Spectral bandwidth and directional spreading are fundamental wave indices for 109 freak wave occurrence. BFI has been considered as a good freak wave occurrence indicator 110 (Janssen, 2003), yet it does not work very well for directional ocean waves (Gramstad and Trulsen, 111 2007; Onorato et al., 2010). Steepness in cases 1 to 6 is always above 0.08 when freak waves 112 happen is always above 0.08, which is a relatively large value for ocean waves' statistical 113 characteristics (Fig. 2). Spectral bandwidth is parameterized by frequency peakedness. The 114 temporal change of frequency peakedness (Fig. 3) is often time similar with that of BFI (Fig. 4) 115 for on account of the direct proportion relation between them according to Eq. (2), such as in cases 116 1, 4, 5, and 6. BFIs at freak wave occurrence time are too small to be consistent with experimental 117 and physical conclusions; BFI is supposed to be larger than 1 when freak waves occurs (Janssen, 118 2003). Similar results are also found by Bertotti and Cavaleri (2008), Burgers et al. (2008). Freak 119 waves are influenced significantly by the directionality of ocean waves and it is almost impossible 120 to generate freak waves in large directional spreading. As such, Hence the directionality of ocean 121 waves is thought to be responsible for the inconsistency of BFI values. The directional spreading 122 values among cases 1 to 6 are relatively small-and are : less than 25 °except case2 (37.3 °) (Fig. 5). 123 It also demonstrates that the freak waves are not clearly related to any wave parameter's absolute 124 value. In contrast, the freak waves should be more associated with the wave parameter's value 125 relative to before and after during a period of time.

In summary, there are no obvious relationships between single wave parameters and freak wave incidents. Freak wave is moreare considered as a result ofto result from B-F instability, so it should be triggered under multi-conditions rather than one and it is not easy to find any clues from single wave parameters.

Joint distributions of multi-wave parameters that are in close relation with freak wave 131 occurrence are more reasonable representation. Tamura et al. (2009), and In et al. (2009) have 132 introduced frequency peakedness-directional spreading joint distribution to explore the freak wave 133 occurrence circumstance. The joint distributions of two freak wave samples that they used in their 134 research show similar visual feature. We find that there are always some abrupt changes in 135 frequency peakedness when the peak number of spectrum varieschanges from single-peak to 136 double-peak. For this reason, the frequency peakedness is not used in the joint distributions. Freak 137 waves are strong nonlinear phenomena, whose occurrences are closely related to ocean waves' 138 directionality. With a consideration of nonlinearity and directionality of ocean waves, wave 139 directional spreading-steepness joint distribution is used to analyze the freak wave incidents in this 140 research.

An obvious visual common feature is shown in six wave directional spreading-steepness joint 142 distributions (Fig. 6). Although it is not obvious in any single parameter, the joint distributions 143 show large steepness and small directional spreading characteristics at freak waves' time. This is 144 quantitatively consistent with experimental and theoretical research conclusions (Gramstad and 145 Trulsen, 2007; Waseda et al., 2009; Onorato et al., 2010). Second, the points are intensive around 146 freak waves' time. It means that large steepness and small directional spreading are continuous 147 over a long period of time. New information given in two characteristics implies certain 148 circumstance that is suitable for triggering freak waves. A continuous sea state with large

|----|-------|-------|----|

steepness (>0.08) and small directional spreading (<27 °) lasting a long time means a "freakish" sea state. Third, the freak wave occurrence time is always near or in the extreme point of joint distribution. It demonstrates the freak wave sea states are near or at the maximum of wave steepness or minimum of directional spreading.</p>

The Case 2 was moderate sea state; the steepness was 0.082 and the directional spreading 154 was 37.3 ° when the suspected freak wave occurred. The directional spreading in Case 2 is too 155 broad to trigger freak waves according to experimental and numerical research results. But for 156 local characteristicHowever, it is relatively small during seven days period (Fig.6, Case2). The 157 freak wave occurrence point is also on the upper left corner of Fig. 6, which is similar with 158 distribution in other cases. For this, Hence it is thought that freak waves are dependent more on 159 relative sea states rather than absolute sea states. Some freak wave incidents also occurred in 160 rather low sea states with the scenario of rapidly changing conditions or crossing seas (Toffoli et 161 al., 2004). Joint distribution in Case2 (Fig. 6) shows a rapid change condition-in direction 162 spreading, and therefore itwhich may be responsible for the suspected freak waves. The obvious 163 visual commonness commonality of the joint distribution shows local extreme conditions and 164 rapid changes of sea state parameters. It always signifies a considerable increase of freak wave 165 occurrence as wave steepness increases and directional spreading narrows. What's is more, the a 166 long duration of this combination may be necessary for "freakish" sea states; how long is not clear 167 from the present evidence and awaits future study.

### 168 4 Conclusions

Both experimental Experimental and theoretical approaches both suggest that the freak waves 170 are triggered under small directional spreading, large steepness and narrow spectrum bandwidth 171 conditions. The attempt to characterize freak wave sea states from single wave parameters is likely 172 impossible. The characteristics with regard to variability of steepness and directional spreading are 173 shown by joint distributions. There are regions that always mean "freakish" seas, which are 174 situated on the upper left corner of the joint distribution figure. In long duration joint distribution 175 of directional spreading-steepness, "freakish" sea states have a visual common feature that 176 steepness is large, and directional spreading is relatively narrow relatively and the state lasts a 177 long time.

Multi-dimensional evolution of wave parameters contains more information, so it is better

- suited for more variables analysis. The visual commonness commonality here suggests feature
- would be supposed to be used as a tool to characterize freak wave sea states and can be validated
- by long time-series observation in the future.
- Acknowledgment. We thank the reviewers for their valuable comments. We are grateful to the
- Physical Oceanography Distributed Active Archive Center (PO.DAAC) at the NASA Jet
- Propulsion Laboratory (JPL) for the CCMP wind. This work is supported by the National Natural
- Science Foundation of China (Grant No. 41406032, 41206021)

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

- Caption of figures:
- Figure 1. Time series of simulated significant wave height (case1-case6), redlines refer to the freak
- waves occurrence time.
- Figure 2. Time series of simulated wave steepness (case1-case6), redlines refer to the freak waves
- occurrence time.
- Figure 3. Time series of simulated frequency peakedness (case1-case6), redlines refer to the freak
- waves occurrence time.
- Figure 4. Time series of simulated BFIs (case1-case6), redlines refer to the freak waves occurrence
- time.
- Figure 5. Time series of simulated directional spreading (case1-case6), redlines refer to the freakwaves occurrence time.
- Figure 6. Joint scatter plot of directional spreading and steepness by 1 hour during 7-20 days
- around the freak waves occurrence time (case1-case6), red star refer to the freak wave occurrence
- time, green rectangles refer to the start and end time.
- 255

Figure 1