# Peer review of "Common characteristics of directional spreading-steepness joint 1 distribution in freak wave events 2 3 S. H. Liu1±, Y. Z. Li2, and X. Y. Yue1 4 1National Marine Data & Information Service, Tianjin, China 5 2Applied Ocean Physic and Engineer"

_Ocean Science, 2015_

## Referee Comment (RC1) · Anonymous Referee #1 · 13 Feb 2016

I have reviewed the manuscript [os-2015-93] titled "Common Characteristics of Directional Spreading-Steepness Joint Distribution in Freak Wave Events" by Liu and Yue.

The authors constructed hindcast simulations of ocean conditions to analyze the wave parameters that closely related to freak waves. They found common visual features in direction spreading-steepness joint distributions when freak waves occurred. The common features seem obvious and interesting. The paper is generally written well and easy to follow. I have only minor comments, and suggest that the manuscript be published in OS.

1. The model used in the study is WAVEWATCH III (WW3), which is a spectral model to describe the statistic characteristics of ocean waves. It is impossible to simulate freak

waves using WW3. How to build the connection between the simulated results and the freak waves is important, but unclear in the paper. 2. Page6, line14-17 "Tamura et al., (2009) and In et al., (2009) have introduced frequency peakedness-directional spreading joint distribution". Where "Steepness" is used in the paper instead of "peak- endness". What's the difference between steepness and peakedness in the joint dis- tribution? 3. Page5, Line26, "Burgers et al., (2008)" change to "and Burgers et al. (2008)" 4. Page5, Line 4, "Where" change to "where" 5. Page6, Line 14,"Tamura et al., (2009), In et al., (2009)" change to "Tamura et al. (2009) and In et al. (2009)"

---

## Referee Comment (RC2) · Anonymous Referee #2 · 14 Feb 2016

The study of freak waves is important because of their potential damages to ships, coastal and oceanic structures. However, they are very difficult to observe, which enables the wave model simulation a good choice besides experimental and theoretical approaches. A key study of freak waves is how to predict their occurrence. Usually, this question is studied using some wave parameters derived from wave spectrum. The authors of present study list these parameters, discuss their relationships with freak wave occurrence, find disadvantages of using single parameter and propose their new approach of multi-parameters. The study uses the well-developed third generation wave model—WaveWacth III to simulate wave spectra. The figures show compellent details and the explanations are reasonable. Their result is interesting and meaningful,

giving useful information on future study of freak wave occurrence. The whole paper is well organized with beautiful results and proper discussions. I think it can be published with some modifications.

The following questions are related to this paper:

1. In page 4, lines 4-5: you mentioned that "Short-lived freak waves can last only for 1 to 10 wave periods". I am not quite clear about the expression "wave periods", does it have a value? or how to define this value?

2. In page 4, lines 10-12: in Section 2 "Model configurations", I think it is not clear to readers on the following issues: 1) What is the outer/inner grid (or area) of the model, so you should explain how to define them. 2) You need to explain more about the model setup, i.e. how many source terms are considered in your calculations? 3) The data specification should be more detailed: the resolutions, the time span and your considerations on how to determine the time span (in other words, how do you determine the calculation period of the model).

3. In page 7, lines 1-3: you mentioned that "A continuous sea state with large steepness (> 0.08) and small directional spreading (< 27°) lasting a long time means a "freakish" sea state. Do you have any idea about how long time is enough to generate freak waves?

4. You mentioned three parameters to study freak wave occurrence: steepness, spectra bandwidth and directional spreading. However, you only discussed the joint distribution of steepness and directional spreading. Actually, there are several combinations of these parameters. Why do not you discuss other combinations?

The following are some small suggestions:

1. In page 5: 1) You should explain every parameter in equations 1 and 2, and you missed $\theta$, and the meaning of F is not exact. 2) Line 15: "spectra bandwidth" can be "spectral bandwidth" 3) Line 23: "BFI" should be "BFIs" 4) Line25: "freak waves occurs"
is wrong

2. Both "water waves" and "ocean waves" appear in this paper, I think it is better to use one of them for consistency.

—————————————————————

---

## Author Comment (AC1) · 9 Apr 2016

The response files are attached in the supplement

Please also note the supplement to this comment:
http://www.ocean-sci-discuss.net/os-2015-93/os-2015-93-AC1-supplement.zip

---

## Author Comment (AC2) · 9 Apr 2016

The response files are attached in the supplement

Please also note the supplement to this comment:
http://www.ocean-sci-discuss.net/os-2015-93/os-2015-93-AC2-supplement.zip

---

## Author Response (AR1)

We thank the two anonymous reviewers for their useful comments to our manuscript. We found them valuable that helps to greatly improve the quality of the manuscript. Per your suggestion, we have carefully revised the manuscript (track-changed) following both reviewers' comments. Our point-by-point response (in blue) to the comments are as below:

**Response to reviewer #1.**

I have reviewed the manuscript [os-2015-93] titled "Common Characteristics of Directional Spreading-Steepness Joint Distribution in Freak Wave Events" by Liu and Yue. The authors constructed hindcast simulations of ocean conditions to analyze the wave parameters that closely related to freak waves. They found common visual features in direction spreading-steepness joint distributions when freak waves occurred. The common features seem obvious and interesting. The paper is generally written well and easy to follow. I have only minor comments, and suggest that the manuscript be published in OS.

We thank reviewer #1 for your encouragement. Our point-to-point response to your questions is as below.

1. The model used in the study is WAVEWATCH III (WW3), which is a spectral model to describe the statistic characteristics of ocean waves. It is impossible to simulate freak waves using WW3. How to build the connection between the simulated results and the freak waves is important, but unclear in the paper.

**Response:** We agree with the reviewer that in our previous version there were not enough information on the connection between the simulated results and freak waves. As the reviewer pointed out, it is impossible to simulate freak waves in WW3.

However, the freak wave-related parameters deduced from simulated results can be considered as an approximation of corresponding parameters of statistical sea states which is pertinent to freak waves. Some previous studies (Zakharov et al., 2007; Badulin et al., 2008; Toffoli and Bitner-Gregersen, 2011)have indicated that the freak wave only has a very limited influence on the statistical properties of ocean waves for its typically timescale is only few wave periods, and we have highlighted this in the first paragraph of Model Configurations:

**2 Model configurations**

As a state-of-the-art third generation spectral model, WAVEWATCH III (WW3) (Tolman, 2002, 2009) offers good descriptions of statistical sea states from a kinetic approach that well mimics directional spectrum. Short-lived freak waves can last only for 1 to 10 wave periods (Janssen, 2003) and hardly influence relatively long-time wave statistical characteristics (Toffoli and Bitner-Gregersen, 2011). Nevertheless, even in complex conditions, the evolution of spectrum with the spectral kinetic description appears to be consistent both qualitatively and quantitatively with solutions for the weakly nonlinear dynamical equations for water waves (Zakharov et al., 2007; Badulin et al., 2008).

In summary, our simulated results of sea states do not contain information of freak waves, although they can be used to indicate the conditions when with freak waves occurring. In the first paragraph of Model Configurations, more explanations about freak waves and simulated results are added in order to avoid the confusions.

[Line 59-62, line64, os-2015-93_manuscript_revised_marked.pdf (hereafter, os..pdf)]:

that well mimics directional spectrum. Although the WW3 model can not give the simulation of freak waves, the freak-wave related parameters deduced from simulated results can be considered as an approximation of corresponding parameters of statistical sea states which is pertinent to freak waves. Short-lived freak waves can last only for 1 to 10 a few

Nevertheless, eEven in complex conditions,

2. Page6, line14-17 "Tamura et al., (2009) and In et al., (2009) have introduced frequency peakedness-directional spreading joint distribution". Where "Steepness" is used in the paper instead of "peakendness". What's the difference between steepness and peakedness in the joint distribution?

**Response:** We agree with reviewer that in previous studies, frequency peakedness-directional spreading joint distribution has been introduced (Tamura et al. (2009; In et al., 2009). However, in this paper, we tend to use 'Steepness' instead of 'frequency peakedness'. The parameterization of 'frequency peakedness' in WW3 model adopted the method of Goda (1970), and is different in the parameterization of single-peaked spectrum and double-peaked spectrum. We find that there are always some abrupt changes in frequency peakedness when the spectrum switches from single peak to double peak. Goda's approach to parameterize the frequency-peakedness is not always suited in double-peaked spectrum. So the frequency peakedness is not used in the joint distributions.

Freak waves are nonlinear phenomena essentially and wave steepness is the direct parameter to characterize the nonlinear level of ocean waves. So the steepness is adopted as the joint distribution factor. We also add some more explanations in the

[line135-137, os..pdf]:

samples that they used in their research show similar visual feature. We find that there are always some abrupt changes in frequency peakedness when the peak number of spectrum varies. For this, the frequency peakedness is not used in the joint distributions. Freak waves are

3. Page5, Line26, "Burgers et al., (2008)" change to "and Burgers et al. (2008)"

**Response:**

We have corrected it according to reviewer's comment. (line119, os..pdf)

4. Page5, Line 4, "Where" change to "where"

**Response:**

We have corrected it according to reviewer's comment. (line99, os..pdf)

5. Page6, Line 14,"Tamura et al., (2009), In et al., (2009)" change to "Tamura et al. (2009) and In et al. (2009)"

**Response:**

We have corrected it according to reviewer's comment. (line132, os..pdf)

**Response to reviewer 2.**

The study of freak waves is important because of their potential damages to ships, coastal and oceanic structures. However, they are very difficult to observe, which enables the wave model simulation a good choice besides experimental and theoretical approaches. A key study of freak waves is how to predict their occurrence.

Usually, this question is studied using some wave parameters derived from wave
spectrum. The authors of present study list these parameters, discuss their
relationships with freak wave occurrence, find disadvantages of using single
parameter and propose their new approach of multi-parameters. The study uses the
well-developed third generation wave modelăˇAˇTWaveWacth III to simulate wave
spectra. The figures show compellent details and the explanations are reasonable.
Their result is interesting and meaningful, giving useful information on future study
of freak wave occurrence. The whole paper is well organized with beautiful results
and proper discussions. I think it can be published with some modifications.
We thank the reviewer for the encouragement. We have modified the manuscript
according to your suggestions and hope you find the revision acceptable.
1. In page 4, lines 4-5: you mentioned that "Short-lived freak waves can last only
for 1 to 10 wave periods". I am not quite clear about the expression "wave periods",
does it have a value? or how to define this value?
**Response:** The "wave periods" is a description about the typical timescale of
freak waves following Janssen (2003). It doesn't have a decisive value and is just a
general description. The periods of ocean waves are always ranging from 1 to 20
seconds. To avoid confusion, we have reworded '1~10 wave periods' to 'a few wave
periods'.
2. In page 4, lines 10-12: in Section 2 "Model configurations", I think it is not
clear to readers on the following issues: 1) What is the outer/inner grid (or area) of the
model, so you should explain how to define them. 2) You need to explain more about
the model setup, i.e. how many source terms are considered in your calculations? 3)
The data specification should be more detailed: the resolutions, the time span and
your considerations on how to determine the time span (in other words, how do you
determine the calculation period of the model).
**Response:** We thank the reviewer for bringing those to our attention.
1) The outer/inner grid information has now been added in Table 1. The
simulated results are easily affected by the errors propagated from the outside
boundary of model grid, so the inner grids that cover the freak wave incidents'
positions are set in the middle of outer grids.
**[line68-72, line86-88, os..pdf]:**
Seven freak wave incidents in the ocean used in this study and the defined model grid
are shown in Table 1. Hindcast simulations are conducted by WW3 multi-grid
technique. The simulated results are easily affected by the errors propagated from the
outside boundary of model grid, so the inner grids that cover the freak wave incidents'
positions are set in the middle of outer grids. The coarse reso-
2) The implementations of WW3 in our simulations follows Tolman (2002,
2009) with few exceptions. As such, we feel it only necessary to a few explanations of
the model setup. The wave directions are set to 36 (by 10 degree), and the number of
frequencies ranges from 0.0412 to 0.4056 is set to 25 levels, with the increment factor
of 1.1. The freak wave incidents do not occur in the shallow water, so only three
source terms are considered in the model: wind-wave interaction term, nonlinear
wave-wave interactions term and a dissipation (whitecapping) term

**[ line73-78, os..pdf].**

**3)** We use the Cross-Calibrated, Multi-Platform Ocean Surface Wind Velocity (Atlas et al., 2011) to force the wave model, which is 0.25 degree resolution at 6 hours interval. A reanalysis ocean current from National Marine Data & Information

Service (China) is also taken into account in the model for the diagnosis of the results.

**[line79-82, os..pdf]**

For WW3, it always needs 1 to 2 days to spin-up the model in cold start conditions. In our simulations, we allow more than 3 days for the model to spin-up before the freak wave incident time. **[line83-85, os..pdf]**

**Table 1.** Time, position information and model set up of freak wave incidents.

| Case | Time(UTC) | Position | Outer grid of model | Inner grid of model | Note |
|------|-----------|----------|---------------------|---------------------|------|
| Case1 | 30 Dec 1980 05:30 | 156º11'E, 31º N | 115º-180ºE, 10º-65ºN | 140º-160ºE, 25º-40ºN | Northwest Pacific |
| Case2 | 23 Jun 2008 04:00 | 144º-145ºE, 35º-36ºN | 115º-180ºE, 10º-65ºN | 140º-160ºE, 25º-40ºN | Northwest Pacific |
| Case3 | 13 Dec 1978 00:00 | 44ºN, 24ºE | 70ºW-10ºE,10º-75ºN | 30ºW-20ºW, 40º-50ºN | Atlantic |
| Case4 | 1 Jan 1995 15:20 | 2º28'E, 58º11'N | 70ºW-10ºE,10º-75ºN | 5ºW-5ºE, 55º-65ºN | New Year Wave |
| Case5.1 | 18 Nov 1997 01:10 | 1º44'E, 60º45'N | 70ºW-10ºE,10º-75ºN | 5ºW-5ºE, 55º-65ºN | Alwyn oil platform |
| Case5.2 | 20 Nov 1997 01:51 | | | | |
| Case6 | 27 Jul 2002 12:00 | 22.17ºE, 37.97 ºS | 0.5 ºE-60ºE, 70ºS-0ºN | 17ºE-27ºE, 43ºS-33ºS | FA platform |

case1, case2 and case3 are for ship sinkings which are thought to be caused by freak waves.

case4, case5 and case6 are freak waves that are recorded by in-situ measurements.

3. In page 7, lines 1-3: you mentioned that "A continuous sea state with large steepness ($> 0.08$) and small directional spreading ($< 27°$) lasting a long time means a

"freakish" sea state. Do you have any idea about how long time is enough to generate freak waves?

**Response:** The conclusion is deduced from the seven freak wave incidents. It means that "the mentioned conditions" is easy to generate freak waves, although it's not a sufficient condition. We don't have a clear answer about how long time is enough to generate freak waves and will keep this in mind in the future study.

4. You mentioned three parameters to study freak wave occurrence: steepness, spectra bandwidth and directional spreading. However, you only discussed the joint distribution of steepness and directional spreading. Actually, there are several combinations of these parameters. Why do not you discuss other combinations?

**Response:** As the response to reviewer 1 in the 2[nd] question, the parameterization of frequency peakedness is not always performance well in the double peaked spectrum. So we used steepness instead of frequency peakedness. **[line 135-137,**

**os..pdf]**
The following are some small suggestions:
1. In page 5: 1) You should explain every parameter in equations 1 and 2, and you
missed $\theta$, and the meaning of F is not exact. 2) Line 15: "spectra bandwidth" can be
"spectral bandwidth" 3) Line 23: "BFI" should be "BFIs" 4) Line25: "freak waves
occurs" is wrong
**Response:**
1) More explanations about $\sigma, \theta$ and *F* are added.
$\sigma$ is the relative radian frequency, $\theta$ is the wave direction, *F* is the wave energy
density spectrum,
**(line99-100, os..pdf)**
2) "spectra bandwidth" is revised to "spectral bandwidth" **(line93; line109,114,**
**os..pdf)**
3) "BFI" revised to "BFIs" **(line116,247, os..pdf)**
2. Both "water waves" and "ocean waves" appear in this paper, I think it is better to
use one of them for consistency.
**Response:** "water waves" is revised to "ocean waves" **(line31,34,66,120,**
**os..pdf).**
We thank review #2 for your useful comments that improves the quality of the
manuscript.
**We also make some other modifications or corrections when checking the**
**manuscript. (Line3-7,180-181(authors are modified to the first version of**
**manuscript);line158)**

[revised manuscript text omitted]

                          Figure 1

[Figure]

                          Figure 2

[Figure]

                          Figure 3

[Figure]

                                          Figure 4

[Figure]

                                          Figure 5

[Figure]

                                          Figure 6

---

## Author Response (AR2)

Thanks so much for the editor's comments and corrections of grammar errors in the manuscript. Those are all valuable and very helpful for revising and improving our paper. The main corrections in the paper and the responds to the comments are as flowing:

**Comments to the Author:**

Thank-you for your revised manuscript. Now I am asking for technical corrections, mainly to clarify the meaning but in one case (end of section 3) to include your response about "how long" in the paper.

Line 13. "occurring . ."

Accepted (line13)

Lines 16-17. ". . Besides large steepness and small directional spreading, a long . ." Accepted. (line16-17)

Line 18. Omit "the sequential variation of"?

Accepted. (line18-19)

Line 19. Omit "continuous"?

It's better to keep "continuous". (line19)

Line 20. Maybe "termed" (meaning "called") rather than "supposed to be"

This sentence aim to express that "sea states of local large steepness and small directional spreading are suitable for the generation of freak waves, but those conditions are not sufficient. We just suppose the sea states are "freakish" sea states" "be "freakish" sea states" is revised to "generate freak waves" (line20-21)

Line 28. "to be" not "as"

Accepted. (line29)

Line 33. Omit "more".

Accepted. (line34)

Line 37. ". . and extreme . ."

Accepted. (line38)

Line 38. "various" not "different".

Accepted. (line39)

Line 40. Omit "Its"

Accepted. (line41)

Line 51. "findings" not "achievements"

Accepted. (line52)

Line 52. "freak waves. This check gives useful information . ."

Accepted. (line53)

Line 57. "mimics the directional spectrum. . . can not simulate freak"

Accepted. (line58)

Line 62. ". . evolution of the spectrum . .". I do not understand "spectral kinetic description"

"spectral kinetic description" should be "kinetic description"; "with" revised to "within" (line63)

Lines 71-72. ". . The wave directions are resolved to  $10^{\circ}$  (36 "bins") and frequencies range from 0.0412 to 0.4056 in 25 bands, with the . ."

**Accepted. (line73-75)**

Lines 75-77. "We force the wave model using the Cross-Calibrated, Multi-Platform Ocean Surface Wind Velocity (Atlas et al., 2011), which has 0.25 degree resolution . ."

Accepted. (line77-79)

Lines 80-81. ". . Tolman, 2002). WW3 always needs 1 to 2 days to spin-up in "cold start" conditions. In our simulations, we allow . ."

Accepted. (line82-83)

Line 92. ". . parameters having a close relationship . ."

Accepted. (line94)

Line 96. Omit "relative"?

"relative" is more precise (line98)

Line 98. ". . takes a local"

Accepted. (line100)

Line 99. ". . near-extreme . ."

Accepted. (line101)

Line 102. ". . indicates that freak wave events may occur when"

Accepted. (line104)

Line 103. ". . series, unlike the other cases' quasi . ."

Accepted. (line105)

Lines 109-110. ". . . Steepness in cases 1 to 6 when freak waves happen is always above 0.08, which . ."

Accepted. (line111-112)

Line 112. "... BFI (Fig. 4) on account of the direct proportion relation"

Accepted. (line115)

Line 113. ". . (2), as in cases . ."

Accepted. (line115)

Line 118. ". . spreading. Hence the directionality . ."

Accepted. (line120)

Lines 119-120. ". . inconsistency of BFI values. The directional spreading . . relatively small: less than . ."

Accepted. (line121-122)

Line 124. "wave incidents. Freak waves are considered to result from B-F instability, so should be"

Accepted. (line127)

Line 127. Omit "multi-"

Accepted. (line130)

Line 128. ". . (2009) and In et al. . ."

Accepted. (line131)

Line 132. "frequency peakedness when the spectrum changes from single-peak to double-peak. For this reason, the frequency . ."

Accepted. (line135-136)

Lines 151-152. Omit "for local characteristic"? Then ". . results. However, it is relatively small . ."

**Accepted. (line155-156)**

Line 154. "other cases. Hence it is thought . ." Accepted. (line158)

Lines 157-158. ". . rapid change in direction spreading, which may be responsible . . obvious visual commonality of the joint"

Accepted. (line161-163)

Lines 161-162. ". . What is more, a long duration . . "freakish" sea states; how long is not clear from the present evidence and awaits future study."

Accepted. (line165-167)

Line 164. "Experimental and theoretical approaches both suggest . ."

Accepted. (line169)

Lines 170-171. ". .is large, directional spreading is relatively narrow and the state lasts a long time." (line176)

Accepted.

Line 173. Omit "more variables"?

Accepted. (line179)

Lines 173-174. ". . The visual commonality here suggests a tool to characterize . ." Accepted. (line179-180)

Some more corrections:

1. "Liu1\*" revised to "Liu1" (line3)

2. Omit "41206021" (line 185)